# Modification of Vip3Ab1 C-Terminus Confers Broadened Plant Protection from Lepidopteran Pests

**DOI:** 10.3390/toxins11060316

**Published:** 2019-06-03

**Authors:** Megan S. Sopko, Kenneth E. Narva, Andrew J. Bowling, Heather E. Pence, James J. Hasler, Theodore J. Letherer, Cory M. Larsen, Marc D. Zack

**Affiliations:** Corteva Agriscience, Indianapolis, IN 46268, USA; knarva@greenlightbio.com (K.E.N.); andrew.bowling@corteva.com (A.J.B.); heather.pence@corteva.com (H.E.P.); james.hasler@corteva.com (J.J.H.); ted.letherer@corteva.com (T.J.L.); cory.larsen@corteva.com (C.M.L.)

**Keywords:** vegetative insecticidal protein, Vip3, spodoptera, lepidoptera

## Abstract

Vegetative insecticidal proteins (Vips) from *Bacillus thuringiensis* (Bt) are unique from crystal (Cry) proteins found in Bt parasporal inclusions as they are secreted during the bacterial vegetative growth phase and bind unique receptors to exert their insecticidal effects. We previously demonstrated that large modifications of the Vip3 C-terminus could redirect insecticidal spectrum but results in an unstable protein with no lethal activity. In the present work, we have generated a new Vip3 protein, Vip3Ab1-740, via modest modification of the Vip3Ab1 C-terminus. Vip3Ab1-740 is readily processed by midgut fluid enzymes and has lethal activity towards *Spodoptera eridania*, which is not observed with the Vip3Ab1 parent protein. Importantly, Vip3Ab1-740 does retain the lethal activity of Vip3Ab1 against other important lepidopteran pests. Furthermore, transgenic plants expressing Vip3Ab1-740 are protected against *S. eridania*, *Spodoptera frugiperda*, *Helicoverpa zea*, and *Pseudoplusia includens*. Thus, these studies demonstrate successful engineering of Vip3 proteins at the C-terminus to broaden insecticidal spectrum, which can be employed for functional expression in planta.

## 1. Introduction

Vegetative insecticidal proteins (Vips) have earned an important role in modern agriculture due to their ability to control lepidopteran pests that damage crops and impact yield. As their name indicates, Vip proteins are produced during the vegetative growth phase of *Bacillus thuringiensis* (Bt), the same soil dwelling bacterium that produce delta endotoxins found in crystalline parasporal inclusions, known as Cry proteins. Unlike Cry proteins, Vip insecticidal proteins are secreted from Bt cells and members of the Vip3A subfamily demonstrate potent activity on Cry resistant insects [1,2,3,4]. Thus, Vip3A proteins represent a mode of action different to Cry proteins in commercial cotton and corn crops [5]. Since their identification in 1996 [6], Vip3 research has focused on the characterization of novel Vip3 proteins and their function [7]. The Crickmore Bt database lists 14 members of the Vip3 family with ten proteins comprising the Vip3A subfamily, three Vip3B proteins, and one protein representing the Vip3C family at less than or equal to 95% sequence identity [8]. The overwhelming majority of investigative work has been performed on the Vip3A subfamily as these proteins have demonstrated potent activity against spodopteran insects, a genus found to be less susceptible to several Cry proteins. Thus, this family could be valuable for the development of stacked insecticidal traits.

It is not surprising that the characterization of a multitude of Vip3A proteins has revealed differences in insecticidal spectrum despite a relatively high degree of sequence homology. The sequence diversity found within the Vip3A family is biased towards the C-terminus. Our group has previously demonstrated that exchange of the C-terminal 589 amino acids of Vip3Bc1 with the corresponding region of Vip3Ab1 directs insecticidal activity towards Vip3Ab1 target insects [9]. However, there was a notable loss of lethal activity suggesting that these chimeric proteins may lack the ability to protect plants at low levels of expression. The aim of this study was to investigate the effects of modest C-terminal diversification on the spectrum of Vip3Ab1 and investigate the utility of Vip3A protein engineering in planta.

The insecticidal spectrum of Vip3Ab1 is active on a wide range of lepidopteran insects including *Spodoptera frugiperda* but lacks activity on *Spodoptera eridania*. Here, we generated a novel chimera using the final 177 amino acids from Vip3Ai1 as donor sequence due to its divergence from the Vip3Ab1 and Vip3Aa16 C-terminal regions. We found that changes in the Vip3Ab1 C-terminus expanded the insecticidal spectrum to include potent activity on *S. eridania* while maintaining activity on *Helicoverpa zea*, *S. frugiperda*, and *Pseudoplusia includens*. Thus, the broadened spectrum of this chimeric protein we call Vip3Ab1-740, demonstrated superior in vitro spectrum relative to the native Vip3Ab1. Further, we generated transgenic *Arabidopsis thaliana* plants that demonstrate the functional utility of Vip3Ab1-740 to protect plants from *S. eridania* feeding in planta. Taken together this work provides a foundation for engineering of Vip3A proteins, through C-terminal modification, towards key lepidopteran pests. In this case, a functional gain of insecticidal activity against a major South American soybean pest, *S. eridania*, is presented.

## 2. Results

### 2.1. Generation of a Vip3Ab1 Chimera

A novel Vip3-based chimeric insecticidal protein, Vip3Ab1-740, was created by combining the N-terminal 612 amino acids of Vip3Ab1 with the C-terminal 177 amino acids of Vip3Ai1, a naturally occurring Vip3A not active against *S. frugiperda*, *S. eridania*, *H. zea*, or *P. includens* in a whole cell diet bioassay in earlier stages of Vip3 screening. A schematic of the design of Vip3Ab1-740 from Vip3Ab1 and Vip3Ai1 is displayed in Figure 1. Utilizing Vip3Ai1 as the reference for amino acid numbering, the proteins are divided into three regions to highlight the known serine protease-processing site of Vip3Ab1 at KVKK^200^. It is worth noting that there is a substitution at this site resulting in KVKN^200^ within the native Vip3Ai1 along with a two amino acid signal sequence extension. The full sequence alignment of the three proteins is displayed in Appendix A. Vip3Ab1-740 was created by combining the highly conserved regions 1 and 2 of Vip3Ab1 with the less conserved region 3 of Vip3Ai1 (Figure 1). The third region was denoted as the point at which the Vip3Ab1 and Vip3Ai1 sequences were overtly divergent, as these proteins only share 57.6% identity within this region. Overall, the Vip3Ab1-740 chimera shares 89.9% overall identity with Vip3Ab1 and 94.8% identity with Vip3Ai1.

### 2.2. Vip3Ab1-740 Midgut Fluid Processing

Vip3Ab1 and Vip3Ab1-740 proteins were both expressed and purified by conventional chromatography as soluble proteins. N-terminal sequencing of Vip3Ab1-740 determined the N-terminus to be ^2^ANMNN, which is expected as the N-terminal methionine has been removed. However, it should be noted that Vip3Ab1-740 contains an additional alanine introduced during cloning, which does not impact activity as this is part of the signal sequence known to be enzymatically removed during activation [6]. We next performed in vitro digestion utilizing *H. zea* and *P. includens* gut fluids to identify any effects of the new C-terminus on the major processing site at KVKK^199^. As expected, digestion with gut fluid from either species produced a major band of approximately 60 kDa with the N-terminus ^200^DSSPA, which is identical to the processing site identified in Vip3Ab1. While this processing site was obvious as early as 30 min under conditions stated in materials and methods, overnight digestion indicated additional C-terminal processing of the 60-kDa fragment as the ^200^DSSPA N-terminus was apparent by SDS-PAGE at lower molecular weights (Figure 2).

### 2.3. Vip3Ab1-740 Chimera In Vitro Bioassay

Next, we investigated the functionality of Vip3Ab1-740 by a diet overlay bioassay. Table 1 contains a comparison of insecticidal activity against key lepidopteran species. Both Vip3Ab1 and the Vip3Ab1-740 chimera had lethal activity against *H. zea*, *S. frugiperda* and *P. includens*. However, only Vip3Ab1-740 had lethal activity against *S. eridania*, while Vip3Ab1 showed no lethal or morbid effect on S. eridania at concentrations as high as 9000 ng/cm^2^. Interestingly, Vip3Ab1-740 retained activity against *P. includens*, however, there was a marked decrease in potency against this insect.

### 2.4. Effects of Co-Feeding of Vip3Ab1 and Vip3Ab1-740 on S. eridania

As we expected Vip3Ab1 and Vip3Ab1-740 to bind the same receptors in vivo, we performed dose-response experiments with *S. eridania* and a serial dilution of Vip3Ab1-740 in the presence of the highest possible concentration of Vip3Ab1 (i.e., the solubility limit of this protein, approximately 1 mg/mL) to determine if Vip3Ab1 could antagonize Vip3Ab1-740 effects in vivo. However, co-feeding Vip3Ab1 with Vip3Ab1-740 resulted in an apparent increase in practical mortality, which accounts for the total number of dead and moribund insects (Figure 3). Thus, the addition of Vip3Ab1 does not antagonize the activity of Vip3Ab1-740, even at concentrations exceeding 1600-fold molar excess as co-feeding treatment at 12 ng/cm^2^ Vip3Ab1-740 also contains Vip3Ab1 at 20,000 ng/cm^2^ (see Materials and Methods).

### 2.5. Activity of Vip3Ab1 and Vip3Ab1-740 in Arabidopsis Bioassay

We evaluated the functional efficacy of Vip3Ab1-740 as a plant incorporated insecticidal protein. We produced single copy transgenic Arabidopsis plants expressing Vip3Ab1-740 under the control of multiple regulatory elements to produce a range of protein levels and compared insect leaf feeding damage to the positive control Vip3Ab1 homozygous transgenic plants. These regulatory element pairs included Arabidopsis Ubiquitin 10 for high expression, Arabidopsis Ubiquitin 3 for medium expression, and Arabidopsis Actin 2 for low expression. Protein quantitation was performed by Western blotting, which also permitted visualization and monitoring of protein integrity.

As expected, the three different regulatory elements produced a range of protein concentrations (Figure 4 and Figure 5) in Arabidopsis leaves as up to 3.10 ng per µg of total soluble protein (TSP) was extracted. There were no signs of Vip3Ab1-740 degradation in the leaves of transgenic plants as antibodies detected a single band of approximately 85 kDa in leaf extracts, indicating that Vip3Ab1-740 was present in full length form (Appendix A). Although all plants were selected by molecular analysis for the presence of a single copy of construct utilizing RT-PCR probes to herbicide selectable marker, DSM2, two retained transgenic plants did not produce any detectable Vip3Ab1-740 protein. Both of the transgenic events lacking protein were generated using constructs containing the Arabidopsis Ubiquitin 10 regulatory element (Events #4 and #42, Figure 5).

Five Arabidopsis leaf punches from each single copy event, as identified by molecular analysis, were assayed in an insect bioassay against *H. zea*, *S. frugiperda*, *S. eridania*, and *P. includens*. Non-transformed wild type Arabidopsis and a third generation homozygous Vip3Ab1 transgenic event were utilized as negative and positive controls, respectively. After four days of feeding, damage was scored as outlined in material and methods. Of the four insects tested, the *H. zea* and *P. includens* larvae were able to consume nearly 100% of the wild type leaf material, while *S. frugiperda* and *S. eridania* consumed less material (0.82 and 0.61 average scores, respectively). Vip3Ab1 positive control plants were protected from *H. zea*, *S. frugiperda*, and *P. includens* feeding damage, while *S. eridania* produced damage scores similar to wild type controls. On the other hand, Vip3Ab1-740 completely protected against all four insects regardless of regulatory element used. Two Vip3Ab1-740 transgenic events that did not express a detectable protein were omitted from the Arabidopsis Ubiquitin 10 regulatory element group damage analysis. Remarkably, even the lowest levels of the Vip3Ab1-740 production (Arabidopsis Actin 2 regulatory element) were able to protect against *S. eridania* feeding when Vip3Ab1 was not efficacious at 20x higher expression levels (Figure 4 and Figure 5).

## 3. Discussion

Vip3 insecticidal proteins represent a mode of action that is distinct from Bt Cry proteins. Vip3A subfamily members have lethal activity against a broad spectrum of crop damaging lepidopteran pests. Importantly, Vip3A proteins have lethal activity against several *Spodoptera* spp. pests, a genus of insects that pose a growing risk to multiple crops including corn and soybean. In this study, we observed differential susceptibility of *S. frugiperda* and *S. eridania* to Vip3Ab1 as there is a potent lethal activity against *S. frugiperda*, while *S. eridania* is not affected. There are few published studies comparing Vip3 activity against multiple species of *Spodoptera*. In all cases we have identified, Vip3A proteins with lethal activity against one species of *Spodoptera* also demonstrate some degree of lethal activity on the second species tested [10,11,12,13,14,15,16]. Bergamasco et al. provided the only direct comparison of the sensitivity of four different *Spodoptera* pests (*S. albula, S. eridania, S. frugiperda,* and *S. cosmoides*) to Vip3Aa while evaluating the combined effects of Vip3Aa and Cry1Ia [10]. This group reported that all species tested were sensitive to Vip3Aa and, with the exception of *S. eridania*, all insects tested demonstrated enhanced effects with Cry1Ia and Vip3Aa insecticidal proteins. Vip3Ab1 is 84% identical to Vip3Aa1, which has a lethal activity on *S. eridania*. Thus, differential *Spodoptera* activity of Vip3Ab1 is unexpected.

Vip3Ab1 and Vip3Aa1 are more than 93% identical over the first 612 amino acids and only 52% identical over the final 177 amino acids. Therefore, we hypothesized that alterations to the C-terminal 177 amino acids of Vip3Ab1 would impact insecticidal spectrum. We previously generated protein chimeras utilizing sequences from Vip3A and Vip3B families and observed redirection of insecticidal activity, but severe protein destabilization [9]. Others have reported that even single amino acid mutations in the final 100 amino acids of Vip3Af1 could have deleterious impact on protein stability and activity [17]. Therefore, the goal of this study was to generate a synthetic Vip3A protein with C-terminal diversity without negative impact on protein stability. We utilized the C-terminal 177 amino acids from an entirely inactive Vip3A protein, Vip3Ai1. We selected this donor sequence because Vip3Ai1 is relatively unique and less characterized. This stretch of amino acids is only 55.7% identical to Vip3Ab1 and 56.2% identical to Vip3Aa1. We have termed this engineered protein Vip3Ab1-740. Vip3Ab1-740 gained lethal activity against *S. eridania* while maintaining lethal activity against *S. frugiperda*, *H. zea*, and *P. includens*. Fang et al. generated similar size protein chimeras utilizing sequences from Vip3Aa and Vip3Ac [18]. One chimera Vip3AcAa, which utilized the N-terminus of Vip3Ac and the C-terminal 189 amino acids of Vip3Aa, demonstrated novel growth inhibition against *Ostrinia nubilalis*, with potencies against *S. frugiperda*, *H. zea,* and *Bombyx mori* that were similar to Vip3Aa [18]. In this case, Vip3Ab1-740 gained new activity towards *S. eridania*, which is not common to either donor protein.

We investigated the proteolytic processing of Vip3Ab1-740 by SDS-PAGE and observed no obvious impact on the rate of degradation relative to Vip3Ab1. In addition, the N-terminal sequence analysis confirmed that *H. zea* and *P. includens* midgut enzymes processed the chimeric protein at the anticipated KVKK↓DSSP processing site. Banyuls et al. [17] reported that changes in the C-terminal region of Vip3Af1 resulted in a shift in the processing to Lys^179^, which is 21 amino acids upstream of the native lysine rich cleavage site KVKK^200^. This observation was accompanied by decreased activity against *S. frugiperda* and *A. segetum*. Therefore, unlike Banyuls et al. mutant proteins, the amino acids from Vip3Ai1 do not destabilize the Vip3Ab1-740 protein or cause aberrant processing.

While gut fluid processing of native Vip3Ab1 at the corresponding lysine (Lys^179^) has not been observed, the presence of Ala^781^ might be expected to affect stability based on other research employing alanine scanning [17]. In the present study, incorporation of the Vip3Ai1 sequence alters this Ala^781^ to His^781^ in Vip3Ab1-740. Banyuls et al. [17] observed decreased stability and insecticidal function of Vip3Af1 when native His^781^ (based on Vip3Ai1 numbering) was changed to an alanine. Thus, it is possible that the change of Ala^781^ to His^781^ is a configuration that promotes stability in Vip3Ab1-740. Therefore, this may be a site for future investigation as additional synthetic Vip3A proteins are designed.

The mechanism by which Vip3Ab1-740 gains *S. eridania* activity is not evident. The data presented herein suggests that this chimera produces a gain of function rather than a redirection of activity towards a new spectrum of lepidoptera akin to the Vip3A/Vip3B chimeras described previously [9]. Supporting this concept, insecticidal activity was maintained against *P. includens*, but there was decrease in potency of purified protein, which might suggest differential impact of the midgut environment rather than a shift in receptor recognition. In addition, co-feeding Vip3Ab1-740 in the presence of 1600-fold molar excess of Vip3Ab1 did not attenuate Vip3Ab1-740 activity on *S. eridania*, which indicates that Vip3Ab1-740 may have increased target site bioavailability or binds a different receptor in vivo, although the latter seems unlikely. To this point, Vip3Ad, Vip3Ae, and Vip3Af, have been shown to bind the same membrane receptors as Vip3Aa in brush border membrane vesicle binding (BBMV) assays [19]. Furthermore, BBMV from Vip3Aa resistant insects show no differences in the ability to bind the Vip3Aa protein in vitro [20]. While the mechanism of resistance is not completely understood, it does emphasize the observation of other groups that, in the case of Vip3A, there are factors beyond membrane binding that contribute to insecticidal specificity [20,21,22]. As the primary aim of these experiments was to investigate the impact and utility of modest C-terminal modification on insecticidal spectrum, other experiments will focus on the mechanism by which C-terminal modification operates.

Lastly, we have demonstrated the functional utility of Vip3Ab1-740 in Arabidopsis. Our in planta data is congruent with in vitro data, which indicated that Vip3Ab1 would not confer plant protection from *S. eridania*. Importantly, relatively low expression levels of Vip3Ab1-740 were able to completely protect Arabidopsis leaves from feeding by *H. zea, S. frugiperda, S. eridania,* and *P. includens*. Thus, the modification of the Vip3Ab1 C-terminus with the Vip3Ai1 donor sequence demonstrates a potentially valuable gain of function that can be translated to *S. eridania* protection in planta.

## 4. Conclusions

In this work, we have demonstrated for the first time that modification of the C-terminus of the Vip3A protein can confer a lethal activity against previously non-susceptible insects. In this case, our changes have broadened the spectrum of Vip3Ab1 to include *S. eridania*, a major pest of South American soybean crops. Thus, the chimeric protein, Vip3Ab1-740, demonstrates a potent lethal activity against four major pests of corn and soybean, making this gene an attractive candidate for use in crops. In vitro co-feeding of these proteins to *S. eridania* indicated that Vip3Ab1 does not compete with Vip3Ab1-740 in the *S. eridania* midgut and suggests that factors beyond receptor binding (i.e., stability) are likely responsible for the lack of activity. Future experiments will focus on a more precise mechanism for the differential spectrum of Vip3Ab1 and Vip3Ab1-740. The current findings are supported by in planta data, which indicates that Arabidopsis plants expressing Vip3Ab1-740 are protected from damage from *S. eridania* feeding, while plants expressing Vip3Ab1 are susceptible to damage from *S. eridania* feeding. Thus, we have demonstrated that strategies focused on engineering of the Vip3A C-terminus can alter the functional spectrum of plant-expressed proteins.

## 5. Materials and Methods

### 5.1. Gene and Protein Sequences

The sequence for Vip3Ab1 corresponds to GenBank Accession AAR40284.1. DIG740 was identified from strain DBt11861 of our internal strain collection. This sequence was found to be identical to GenBank Accession KC156693.1, which corresponds to Vip3Ai1. All sequences are displayed in Appendix A.

### 5.2. Construct Design for Bacterial Expression

To generate a chimeric Vip gene consisting of the first 1851 bp of Vip3Ab1 and the last 540 bp of DIG740, polymerase chain reactions (PCR) were performed to generate the two products, then a second round of PCR was performed to join the two products using overlapping PCR with the forward Vip3Ab1 primer and the reverse DIG740 primer. Primers are described in Appendix A. PCR was performed using the Phire Hot Start II polymerase (Thermo Fisher Scientific, Waltham, MA, USA) in the following reaction: 27 μL H_2_O, 10 μL 5X Phire buffer, 1 μL dNTP mix, 5 μL Forward (10 μM), 5 μL Reverse (10 μM), 1 μL Vip3Ab1 (20 ng/ μL) and 1 μL Phire polymerase. Cycling was 98 °C/30 s followed by 30 cycles of 98 °C/5 s, 50.8 °C/5 s, 72 °C/2 min followed by a final extension at 72 °C/1 min then hold at 4 °C. For the assembly of the full length chimeric gene the following reaction was used: 26 μL H_2_O, 10 μL 5× Phire buffer, 1 μL dNTP mix, 5 μL Vip3Ab1 Forward (10 μM), 5 μL Vip3Ai1 Reverse (10 μM), 1 μL Part A reaction, 1 μL Part B reaction and 1 μL Phire polymerase. Cycling was as above except using 2.5 min extension time. The 2397 bp product was gel purified and ligated into pCR-BluntII-TOPO (Thermo Fisher Scientific) and sequenced. A clone having the correct sequence was digested with BamHI and the fragment was gel purified. The fragment was ligated into pET24(+) (MilliporeSigma, Burlington, MA, USA) which was linearized with BamHI and rSAP treated using NEB T4 DNA ligase. Minipreps were performed and a clone having the gene in the proper orientation was selected for expression.

### 5.3. Protein Expression and Purification

Vip3Ab1 and Vip3Ab1-740 were initially expressed in *E. coli* BL21 (DE3) cells. Briefly, seed and production cultures were grown in TB media (Fisher BioReagents™ Terrific Broth, Cat#BP9728-2, Thermo Fisher Scientific, Waltham, MA, USA) supplemented with 50 µg/mL of kanamycin (GoldBio, Cat#K-120-100, St. Louis, MO, USA). Production cultures were inoculated with a 1:200 dilution of an overnight seed culture (16–18 h) and incubated at 37 °C, 225 rpm, in a 2.5 cm throw shaker, until mid-log phase (OD_600_ 0.6–0.9). Protein expression was then induced with the addition of 1 mM IPTG and incubated for 18–20 h at 18 °C, 200 rpm in a 2.5 cm throw shaker. Larger scale (>1 L) of both proteins used recombinant *Pseudomonas* fluorescens strains as described previously [23]. Proteins purified from both systems have been shown to have equivalent potency, and the purification methods were the same. Vip3Ab1 was purified as previously described [9], and Vip3Ab1-740 was purified in a similar manner. Harvested cells containing Vip3Ab1-740 were sonicated in lysis buffer consisting of 50 mM Tris-HCl (pH 8.0), 1 M NaCl, 10% glycerol and 2 mM EDTA with 50 µL of protease inhibitor cocktail (Sigma-Aldrich, St. Louis, MO, USA) per 25 mL buffer. The extract was centrifuged at 20,000× *g* for 40 min. The soluble protein in the supernatant was precipitated with 50% ammonium sulfate and centrifuged at 20,000× *g* for 20 min. The pellet was resuspended in 50 mM Tris-HCl (pH 8.0) and purified by anion exchange chromatography using a HiTrap™ Q HP 5 mL column with an AKTA Purifier chromatography system (GE Healthcare, Chicago, IL, USA). The column was equilibrated in 50 mM Tris-HCl (pH 8.0), and proteins were eluted with a stepwise gradient to 1 M NaCl. Protein-containing fractions were combined and concentrated using Amicon^®^ Ultra-15 Centrifugal Filter Devices with a 30 kDa MWCO (MilliporeSigma, Burlington, MA, USA). Proteins were desalted to 50 mM Tris-HCl (pH 8.0) using Zeba^®^ Spin Desalting Columns, 7 MWCO (Thermo Fisher Scientific) or by dialysis in 50 mM Tris-HCl (pH 8.0) using Slide-A-Lyzer^®^ Dialysis Cassettes, 20,000 MWCO (Thermo Fisher Scientific). Total protein concentrations were measured with the NanoDrop 2000C Spectrophotometer (Thermo Fisher Scientific), using the A280 method. Vip3Ai1 was expressed in *P. fluorescens* but was not purified due to lack of activity on insect bioassay.

### 5.4. SDS-PAGE Analysis

SDS-PAGE analysis was performed using NuPAGE^®^ Novex^®^ 4–12% Bis-Tris Protein Gels (Thermo Scientific). Proteins were diluted in a 4X NuPAGE^®^ LDS Sample Buffer (Thermo Fisher Scientific, Waltham, MA, USA) containing 100 mM tris 2-carboxyethyl phosphine (TCEP) reductant prior to loading onto the gel. Ten µL of Novex^®^ Sharp Pre-stained Protein Standard (Thermo Fisher Scientific) was loaded onto one lane of each gel. Gels were run according to the manufacturer’s recommendations using a NuPAGE^®^ MES SDS Running Buffer (Thermo Fisher Scientific) and stained with SimplyBlue™ SafeStain (Thermo Fisher Scientific), then destained in water and imaged on a flatbed scanner.

### 5.5. Lepidopteran Midgut Fluid Protein Digestion and N-Terminal Sequencing

Midgut fluids were prepared from *H. zea* and *P. includens* larvae and normalized by total proteolytic activity using a BODIPY-casein degradation assay as previously described [9]. Prepared midgut fluids were aliquoted and stored at approximately −80 °C. Proteins were added to reactions at 150 µg/mL final concentration. All digestions were performed at both pH 8.0 and pH 10.0. Control reactions were prepared containing no insect gut fluid. Reactions were incubated with shaking at 30 °C for various time intervals, and the protease inhibitor cocktail (Sigma-Aldrich, St. Louis, MO, USA) was added to terminate the reactions prior to the SDS-PAGE analysis. For N-terminal sequencing, protein products were resolved by SDS-PAGE and transferred to a PVDF membrane using an iBlot^®^2 Transfer Stack, PVDF, mini with an iBlot^®^2 Gel Transfer Device (Thermo Fisher Scientific, Waltham, MA, USA), following the manufacturer’s recommended protocol. The PVDF membrane was stained with Coomassie Brilliant Blue R250 Staining Solution (Bio-Rad Laboratories, Hercules, CA, USA) and destained with 45% methanol, 10% acetic acid. The membrane was rinsed with water and allowed to air dry. The target bands were excised and analyzed by Edman degradation using a Shimadzu PPSQ-33A protein sequencer (Shimadzu, Kyoto, Japan). The data was analyzed with the Shimadzu data analysis software.

### 5.6. Insect Diet Overlay Bioassays

Bioassays were conducted in 128-well bioassay trays (C-D International, Pitman, NJ). A 40 µL aliquot of protein sample was delivered onto the surface of multispecies lepidopteran diet (Southland Products, Lake Village, AR, USA) in each well. For co-feeding experiments, Vip3Ab1-740 was diluted directly in Vip3Ab1 protein stocks, which were maintained at a concentration of 1 mg/mL. The final ratios of Vip3Ab1-740 and Vip3Ab1 are shown below. The treated trays were air dried, and one individual *Helicoverpa zea*, *Spodoptera frugiperda, Spodoptera eridania*, or *Pseudoplusia includens* larva (24 to 48 h after eclosion) was deposited on the treated diet surface. The infested wells were then sealed with adhesive sheets of clear plastic vented to allow gas exchange (C-D International). Bioassay trays were held under controlled environmental conditions (28 °C, 40% relative humidity, 16:8 h light/dark) for five days. The total number of insects exposed to each protein sample, the number of dead insects and the number of surviving insects were recorded in all insect bioassays. All bioassay results contained less than 15% average mortality. Larvae that had not developed beyond the first instar and did not respond to mechanical stimulation were considered moribund insects and were included in the percent practical mortality computation. Bioassays were conducted under randomized complete block design and replicated at least twice, with 16 larvae per replicate. A co-dosing bioassay was performed with serial dilutions of Vip3Ab1-740 in Vip3Ab1 at a starting concentration of 1 mg/mL, and final protein concentrations are displayed in Table 2. All statistical analyses were conducted using the JMP software version 12.2 (SAS Institute, Cary, NC, USA). Probit analyses of the pooled mortality and moribund data were used to estimate the 50% lethal concentration (LC50). Binomial distribution analysis was used to generate 95% confidence intervals for comparison of Vip3Ab1 protein co-feeding effects.

### 5.7. Plant Expression Constructs

The plant-optimized Vip3Ab1 and Vip3Ab1-740 genes were cloned into Agrobacterium transformation vectors essentially as previously described [24]. DNA sequences for plant-expressed Vip3Ab1-740 can be found in Appendix A.

### 5.8. Generation of Transgenic Arabidopsis Plants

Arabidopsis ecotype Columbia was used for transformation. Standard Arabidopsis transformation procedure was used to produce transgenic seed by inflorescence dip method [25]. The T1 seeds were sown on selection trays (10.5”x21”x1”, T.O. Plastics Inc., Clearwater, MN, USA). For this 400 mg of cold stratified seeds (0.1% agar + 385 mg/L Liberty for 48 h before sowing) were distributed on selection trays using a modified air driven spray apparatus to distribute 10 mL of seed suspension per selection tray. Trays were covered with humidity domes, marked with seed identifier, and placed in a Conviron with an individual watering tray under each flat. The humidifying dome was removed approximately five days post-sowing. The first watering of selection trays was done using sub-irrigation with Hoagland’s fertilizer at approximately 10–14 days post sowing. Vip3Ab1 transgenic controls were generated in a similar manner utilizing herbicide spray selection. Third generation homozygous plants were identified by molecular analysis and maintained as controls.

To achieve a stringent herbicide selection, in addition to stratification with the herbicide, plants are sprayed with a 60 mg/L of gluphosinate (Liberty 280) seven and nine days post-sowing. Resistant to gluphosinate T1 plants were transplanted from selection trays into 2-inch pots and allowed to grow for 7–10 days before sampling for molecular analysis. Based on results of the molecular analysis a subset of plants with single transgene copy were retained for expression analyses, insect bioassay and seed production.

### 5.9. Molecular Analysis of Transgenic Arabidopsis Plants

For qPCR, approximately 0.5 square centimeter of the Arabidopsis leaf was pinched off each plant and collected in a 96-well DNA extraction plate (#19560, Qiagen, Hilden, Germany). Three hundred uL of extraction buffer was added to each well and tissue was disrupted by bead beating. After tissue maceration, DNA was isolated using a modified protocol for the BioSprint 96 DNA Plant Kit™ (#941558, Qiagen). Detection of the transgene copy number was conducted by quantitative real time PCR (qPCR) using the LightCycler^®^480II instrument (Roche Applied Science, Penzberg, Germany) with hydrolysis probe assays. qPCR reactions were set for each sample (Appendix A). The reaction combined internal reference gene TAFII (HEX) and the selectable marker gene DSM2 (FAM) assays with the TAFII probe used to control for qPCR efficiency. The qPCR reaction mix was as follows: Ten µL of 1× LightCycler^®^480 Probes Master mix (Roche Applied Science, #04707494001) and containing 0.4 µM of each primer and 0.2 µM of each probe. The crossing-point (Cp) scores and the Relative Quant module of the Roche LightCycler^®^480II Software were used to perform the analyses of real time qPCR data. The PCR cycle was 95 °C to 10 min for initial template denaturation, followed by 40 cycles of 95 °C—1 min, 60 °C—40 s, and 72 °C—1 s. Reactions were finished by cooling down to 40 °C for 1 s. The copy number was estimated using the target (DSM2) to reference (TAFII) ratio. Sequence information for primer sequences and probes is shown in Appendix A. Events identified as single copy for the selectable marker DSM2 were retained for the protein detection and insect bioassay.

### 5.10. Protein Analysis of Transgenic Arabidopsis Leaf Tissues

Single copy events were analyzed for protein accumulation by a quantitative Western blot. Leaf samples (two disks totaling 0.565 cm^2^) were extracted by adding two 4.5 mm steel beads (Daisy Outdoor Products, Rogers, AR, USA) and 150 µL of extraction buffer (PBS + 10 µL/mL protease inhibitor cocktail; Sigma-Aldrich, St. Louis, MO, USA) to each tube. The samples were ground for 3 min in a KLECO bead mill (Garcia Machine, Visalia, CA, USA), then 150 µL of the SDS buffer (PBS + 2% SDS; Thermo Fisher Scientific, Waltham, MA, USA) was added to each tube and the samples were rocked at room temperature for 10 min. After rocking, the samples were centrifuged at 3000× *g* for 5 min. The resulting supernatant leaf extracts were then mixed with an appropriate amount of sample buffer (NuPage LDS 4X Sample buffer with 40 mM dithiothreitol; Thermo Fisher Scientific). Serial dilutions of a purified protein positive control (Vip3Ab1-740) were prepared in a sample buffer as above to a concentration range of (8–0.06 ng/µL). Samples were resolved on NuPAGE^®^ Novex^®^ 4–12% Bis-Tris Protein Gels (Thermo Fisher Scientific) with a MOPS running buffer (Thermo Fisher Scientific).

Gels were transferred to nitrocellulose membranes. Post transfer, the membranes were blocked for 1 h at room temperature in a blocking buffer (1% DIFCO milk in PBST; Becton Dickinson, Franklin Lakes, NJ, USA) followed by incubation with primary antibody (monoclonal anti-Vip3Ab1) at a concentration of 1.0 µg/mL in a blocking buffer overnight at 4 °C. Membranes were then washed and incubated for 1 h at room temperature in a detection antibody (Goat anti-mouse Cy3; GE Healthcare LifeSciences, Marlborough, MA, USA) diluted 1:3000 in PBST. After incubation, the membranes were washed extensively with PBST followed by rinsing three more times with PBS. The membranes were kept in the dark until analyzed on the Typhoon Imager (GE Healthcare LifeSciences). The blots were scanned for fluorescence at 350–400 PMT with an excitation wavelength of 532 nm and an emission filter setting of 580 nm BP30. The images were then processed using the ImageQuant TL software (GE Healthcare LifeSciences).

### 5.11. Arabidopsis Feeding Bioassay

Five healthy, single-copy events from each construct were selected for bioassay. Protein expression results were not available before the bioassay plant selection. Four wild type events were chosen as bioassay negative controls. Four plants expressing Vip3Ab1 were selected as positive bioassay controls. The plants were bioassayed at four weeks old with *H. zea*, corn earworm (CEW); *P. includens*, soybean looper (SBL); *S. frugiperda*, fall armyworm (FAW); and *S. eridania*, southern armyworm (SAW) larvae. A standard paper hole punch, 6 mm diameter producing a disk of 28.3 mm^2^ was used to make five leaf discs per plant. These punches were placed on top of 2% water agar in a 96-well plate, one leaf punch per test well. For each test, wild type plant leaf tissues were sampled first to prevent cross contamination of Bt toxins on the leaf edges, followed by the transformed plants. The leaf punch was cleaned thoroughly with ethanol between plants. Once all leaf punches had been sampled, *H. zea* and *S. frugiperda* experiment plates were egg seeded with black head stage eggs, with about 3-5 eggs in each well. The *P. includens* and *S. eridania* plates were infested with neonate larvae, one larva per well. The infested 96-well plates were sealed with a lid and were placed in a transparent box. The bioassay plates were incubated at 28 °C (16:8 h light:dark, 40% RH) for four days prior to scoring the percent leaf damage. The leaf damage scoring scale is reported in Table 3.

## Figures and Tables

**Figure 1 toxins-11-00316-f001:**
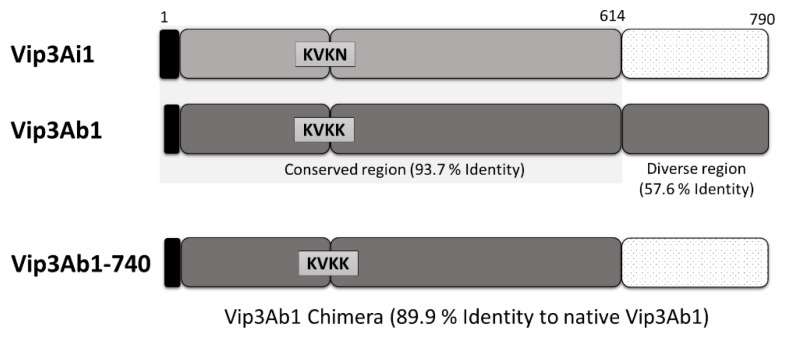
Schematic of the Vip3Ab1-740 chimera protein created by combining the N-terminal 612 amino acids of Vip3Ab1 with the C-terminal 177 amino acids of Vip3Ai1 (DIG740). Amino acid numbering is based on Vip3Ai1, which has an additional two amino acids at the N-terminus relative to Vip3Ab1. Three regions are denoted by rounded boxes. Signal sequences are shown in black.

**Figure 2 toxins-11-00316-f002:**
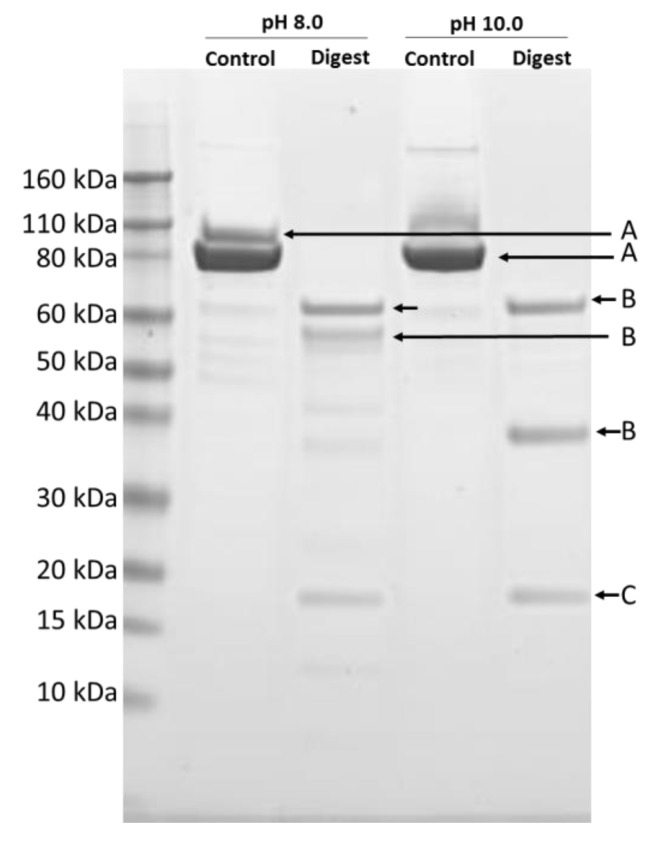
Vip3Ab1-740 was digested overnight at pH 8.0 (left) and 10.0 (right) with *P. includens* midgut enzymes as described in material and methods. Protein products were then resolved by SDS-PAGE and transferred to a polyvinylidene fluoride (PVDF) membrane for N-terminal sequencing. “A” denotes proteins determined to have the N-terminus ^2^ANMNN, bands labeled “B” were found to have the N-terminus at ^200^DSSPA and band “C” is known to be the N-terminal ~20 kDa fragment. Identical results were obtained with *H. zea* midgut enzymes (not shown).

**Figure 3 toxins-11-00316-f003:**
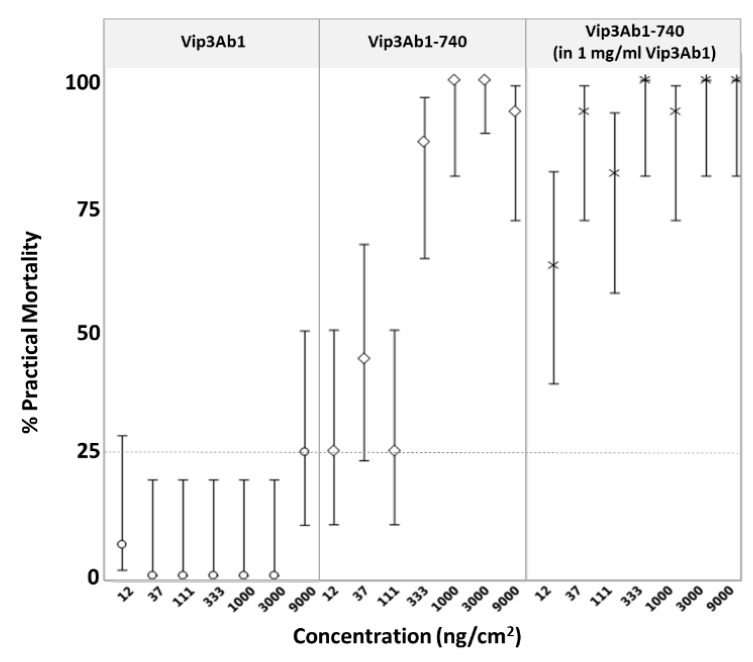
Effects of co-dosing Vip3Ab1 and Vip3Ab1-740 on *S. eridania* using a diet overlay bioassay. Open circles (o) indicate treatment with Vip3Ab1, open diamonds (◊) indicate treatment with Vip3Ab1-740, and (*) indicate co-treatment with several concentrations of Vip3Ab1-740 diluted directly in Vip3Ab1 at 1 mg/mL. Upper confidence limit for buffer treated insects is shown as a dotted line. Practical mortality (%) with associated upper and lower 95% confidence intervals were plotted after five days of exposure.

**Figure 4 toxins-11-00316-f004:**
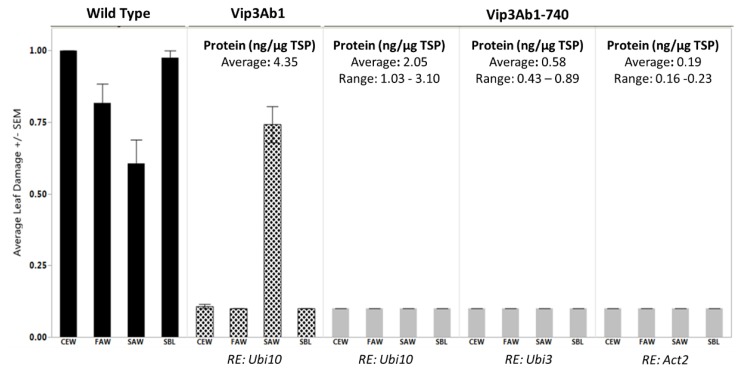
Arabidopsis leaf feeding assay results. Average leaf damage by *H. zea* (Corn Earworm; CEW), *S. frugiperda* (Fall Armyworm; FAW), *S. eridania* (Southern Armyworm; SAW), and *P. includens* (Soybean Looper; SBL). Results represent damage in average of four wild type control plants, four Vip3Ab1 positive control plants, and five independent Vip3Ab1-740 transgenic events under the control of three different regulatory elements (RE). *Post hoc* protein analysis revealed two Arabidopsis Ubi10 events (#4 and #42) lacked the Vip3Ab1-740 protein and were omitted from the analysis. Average Vip3 expression in ng per µg of total soluble protein (TSP) is also provided.

**Figure 5 toxins-11-00316-f005:**
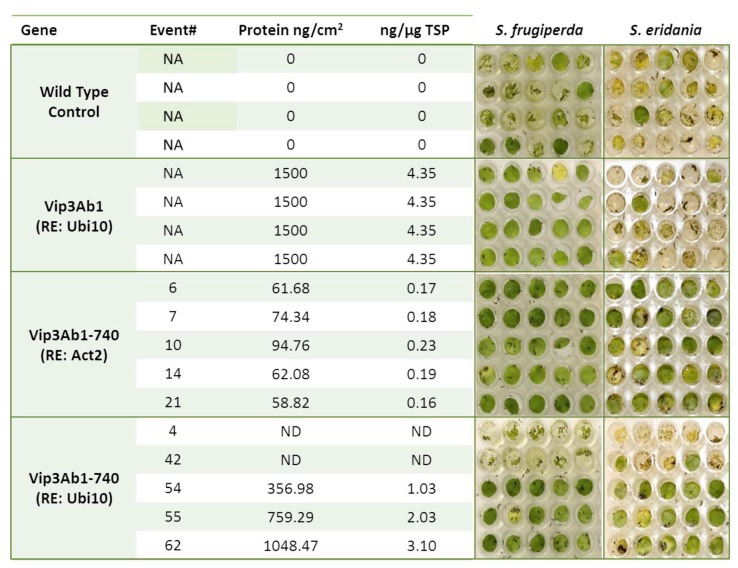
Comparative analysis of *S. frugiperda* and *S. eridania* Arabidopsis leaf feeding assay demonstrating visual differences in leaf protection by Vip3Ab1 and Vip3Ab1-740. Each row contains five leaf punches from individual events as described in Materials and Methods. Columns contain protein analysis both in terms of ng per cm^2^ and ng per µg of total soluble protein (TSP). “NA” indicates Not Applicable and “ND” indicates not detected. Note Arabidopsis Events #4 and #42 did not express Vip3Ab1-740 and were completely consumed by either Spodopteran species.

**Table 1 toxins-11-00316-t001:** Insecticidal activity of Vip3Ab1 and Vip3Ab1-740 against key lepidopteran target insects *H. zea*, *S. frugiperda*, *S. eridania*, and *P. includens*. LC_50_ concentration was calculated using a Probit analysis of the sum of dead and moribund insects relative to the total number of treated insects. “NC” indicates that confidence intervals (CI) were not calculated.

Insect	Vip3Ab1				Vip3Ab1-740			
	LC_50_ (ng/cm^2^)	Lower CI (95%)	Upper CI (95%)	N	LC_50_ (ng/cm^2^)	Lower CI (95%)	Upper CI (95%)	N
***H. zea***	530.6	328.7	851.4	144	912.0	663.7	1262.8	254
***S. frugiperda***	177.9	138.6	230.7	362	80.8	69.0	94.6	1056
***S. eridania***	>9000	NC	NC	256	62.4	46.0	82.8	381
***P. includens***	53.9	37.6	75.1	373	1403.1	1019.2	2005.8	385

**Table 2 toxins-11-00316-t002:** Concentration details for co-dosing bioassay. A serial dilution was performed of Vip3Ab1-740 diluted into Vip3Ab1, and final protein concentrations for each dose as well as the fold molar excess of Vip3Ab1 are displayed.

Vip3Ab1-740 (ng/cm^2^)	Vip3Ab1 (ng/cm^2^)	Molar Excess Vip3Ab1 (fold)
9000.0	11,000	1.2
3000.0	17,000	5.7
1000.0	19,000	19
333.3	19,600	58.8
111.1	19,900	179.1
37.0	19,960	538.9
12.3	19,980	1618.4

**Table 3 toxins-11-00316-t003:** Leaf damage scoring scale for Arabidopsis feeding bioassay. Each score represents a range of percent leaf area damage.

Score	Range of % Leaf Area Damage
0.1	0–10%
0.25	11–25%
0.5	26–50%
0.75	51–75%
0.9	76–90%
1	91–100%

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
