# Peer review of "Modification of Vip3Ab1 C-Terminus Confers Broadened Plant Protection from Lepidopteran Pests"

_toxins, 2019, doi:10.3390/toxins11060316_

Round 1

Reviewer 1 Report

From two Vip3A proteins, one toxic to several Spodoptera species, but not to S. eridiana, and one non toxic to any of them (Vip3Ai), the authors have constructed a chimeric protein, with the N-terminal part of Vip3Ab and the C-terminal 177 amino acids of Vip3Ai. Unexpectedly, the chimeric protein has shown toxicity to S. eridiana, while maintaining the toxicity to the species targeted by Vip3Ab. This “the novo” toxicity is interesting and deserves further study on the biochemical basis.

Major comments:

The supplementary figures lack a legend. Only the title is provided. For example, what is the meaning of the inverted arrow or the different colors in the amino acids in Fig. S1? What is the meaning of Col0 in Fig. S2?

In several parts of the text (i.e., line 62) it is mentioned that Vip3Ai lacks insecticidal activity, but nowhere this is shown or a reference given. This important claim has to be sustained.

The authors do not comment on the important decrease of insecticidal activity to P. includes. While it is correct to say that the chimeric toxin maintains the toxicity to this species, it is not fair not to mention that it has a strong negative effect on the activity. This affects line 219.

The paragraph between lines 210 and 217 is incorrect, since there is a mismatch in the amino acid residues position. The authors have to take into account the differences in the numbering of the Vip3Af1 and Vip3Ab1 proteins. The two positions mentioned (726 and 780) correspond to a highly conserved position in all Vip3A proteins with a Gly, and this does not change in the chimeric protein here produced (residues 729 and 780). The paragraph has to be rewritten.

In line 225, the authors give reference 19 for binding assays. This reference has nothing to do with that. The correct reference would be: Chakroun and Ferré, 2014 Appl. Environ. Microbiol. 2014, 80(20):6258. DOI: 10.1128/AEM.01521-14.

Other comments:

Line 63: The numbering used is not the one of Vip3Ab, but from Vip3Ai, according to the sequence in Figure S1.

Line 80: “which is identical to that of…”. This is not correct, according to Fig. S1. There is an A in Vip3Ab-740 that does not appear in the Vip3Ab sequence.

Line 80 and many more: Species names have to go in italics.

Line 83: Change specie for species.

L 112: Define “practical mortality” since it is here the first time it appears.

L115: The sentence of “contains a dose of Vip3Ab1 at 20,000 ng/cm2” is difficult to understand, Maybe it would be more clear if it was changed to “contains a contribution of Vip3Ab1 of 20,000 ng/cm2”.

Fig. 3: The labels are incorrect. The 3 is missing in the name of the toxins. Also, for the third column, since the concentration at the bottom refers only to the chimeric protein, it would be more correct to label it as “Vip3Ab1-740 (in 1 mg/ml Vip3Ab1)”.

Line 138 and Fig. 5: When referring to event 4, there are two events with the same number in Fig. 5, one in the WT control and another in Ubi10. I do not understand why the WT control has 4 different events. Shouldn’t it say NA, such as in  the positive control?

Legend to Fig. 4: The common name of the insect species is written in different styles, with capital letter the first word, the two words, or both in small letters. Please use the correct one and normalize the nomenclature.

Fig. 5: It is not clear the meaning of columns Protein ng/cm2 and ng/ul TSP. Also, spell out Na and TSP in the legend.

L 204: Substitute (2018) by [17]

L 207: The proteins in Banyuls et al. were not chimeric, but Ala substitution mutants. Change chimeric by mutant.

The Discussion section does not mention the interesting, and unexpected, effect of Vip3Ab on the co-feeding experiments. This deserves some comments. The explanation given in Conclusions (lines246-248) is very ambiguous.

L302-303: Why Vip3Ai was expressed if nothing was done with it?

L 307: Define TCEP. Is it a protease inhibitor?

L 347: Is the title of the heading correct? It has no sense.

L 350: Supplemental Table 2 is not a table, but a figure.

L 371: The most widely used nomenclature for quantitative PCR is not QPCR, but qPCR. Please change.

L 389: Capitalize Western.

L 397: Dithiothreitol has to go with small letter.

L 410: Correct wavelength.

Author Response

Thank you for your helpful suggestions and insight. We have addressed all suggestions in a stepwise fashion below.

Comments and Suggestions for Authors

From two Vip3A proteins, one toxic to several Spodoptera species, but not to S. eridiana, and one non toxic to any of them (Vip3Ai), the authors have constructed a chimeric protein, with the N-terminal part of Vip3Ab and the C-terminal 177 amino acids of Vip3Ai. Unexpectedly, the chimeric protein has shown toxicity to S. eridiana, while maintaining the toxicity to the species targeted by Vip3Ab. This “the novo” toxicity is interesting and deserves further study on the biochemical basis.

Major comments:

The supplementary figures lack a legend. Only the title is provided. For example, what is the meaning of the inverted arrow or the different colors in the amino acids in Fig. S1? What is the meaning of Col0 in Fig. S2?

Thank you for highlighting this issue. Due to the journals request to include figures as part of the text for submission, the supplementary figure legends were inadvertently removed.  We have now included them with the manuscript resubmission. 

In several parts of the text (i.e., line 62) it is mentioned that Vip3Ai lacks insecticidal activity, but nowhere this is shown or a reference given. This important claim has to be sustained.

This is an important point that was not expanded within the text.  Similar to other groups, whole cell transformants are pre-screened in diet overlay bioassay and expression is confirmed by PAGE analysis.  However, if there is no activity observed, no effort is made to purify the protein.  We have made adjustments to the text to highlight this point.  We have also included that DIG740 (Vip3Ai1) was screened against a selected group of insects.

The authors do not comment on the important decrease of insecticidal activity to P. includes. While it is correct to say that the chimeric toxin maintains the toxicity to this species, it is not fair not to mention that it has a strong negative effect on the activity. This affects line 219.

This is also an important observation that should be addressed.  We have added text to bring forward this apparent loss of potency by in vitro overlay bioassay.  Importantly, the functional ability to protect plants from P. includens feeding is maintained.

The paragraph between lines 210 and 217 is incorrect, since there is a mismatch in the amino acid residues position. The authors have to take into account the differences in the numbering of the Vip3Af1 and Vip3Ab1 proteins. The two positions mentioned (726 and 780) correspond to a highly conserved position in all Vip3A proteins with a Gly, and this does not change in the chimeric protein here produced (residues 729 and 780). The paragraph has to be rewritten.

Thank you for identifying this error. This has been corrected within the suggested paragraph.  Any inferences regarding amino acid 729 have been omitted as the reviewer correctly indicated that this position is indeed highly conserved.  We have re-written this paragraph to draw attention to the work by Banyuls et al. that may suggest mutations of the C-terminus can destabilize the protein.  In particular, the creation of Vip3Ab1-740 does introduce a HIS at position 781 (based on Vip3Ai1 numbering).  Mutation of this HIS to an ALA by Banyuls et al. did result in decreased insecticidal activity.  This paragraph has been overhauled to reinforce this reference.

In line 225, the authors give reference 19 for binding assays. This reference has nothing to do with that. The correct reference would be: Chakroun and Ferré, 2014 Appl. Environ. Microbiol. 2014, 80(20):6258. DOI: 10.1128/AEM.01521-14.

This has been corrected.  Thank you for identifying this error.

Other comments:

Line 63: The numbering used is not the one of Vip3Ab, but from Vip3Ai, according to the sequence in Figure S1.

This has been corrected.  Thank you for identifying this error.

Line 80: “which is identical to that of…”. This is not correct, according to Fig. S1. There is an A in Vip3Ab-740 that does not appear in the Vip3Ab sequence.

We have drawn attention to this cloning fingerprint as it has been rephrased in the text.  Early versions of the Vip3Ab1 gene contained this addition due to older cloning techniques (referenced in US patent US2012/0317682A1).  This addition facilitated cloning for plant expression as these constructs with appropriate RE’s were readily available.  This alanine has no impact on Vip3A activity as early production of Vip3Ab1 also contained the same Ala with no effect on activity as it is readily removed by insect protease during the activation process (highlighted by Estruch et al 1996 and again by Zack et al 2017).  Thus, while we draw attention to this important point, we have attempted to be as concise as possible within the text to avoid reader distraction.

Line 80 and many more: Species names have to go in italics.

We have fixed this as it was formatted incorrectly when placed in the Toxins template.

Line 83: Change specie for species.

We have fixed this oversight.

L 112: Define “practical mortality” since it is here the first time it appears.

Thank you.  We have addressed this in the text.

L115: The sentence of “contains a dose of Vip3Ab1 at 20,000 ng/cm2” is difficult to understand, Maybe it would be more clear if it was changed to “contains a contribution of Vip3Ab1 of 20,000 ng/cm2”.

Yes – we agree that this is a better way to state this.  The Authors struggled with the best way to state these results and appreciate this suggestion.  It has been changed.

Fig. 3: The labels are incorrect. The 3 is missing in the name of the toxins. Also, for the third column, since the concentration at the bottom refers only to the chimeric protein, it would be more correct to label it as “Vip3Ab1-740 (in 1 mg/ml Vip3Ab1)”.

We have made these changes in Figure 3.

Line 138 and Fig. 5: When referring to event 4, there are two events with the same number in Fig. 5, one in the WT control and another in Ubi10. I do not understand why the WT control has 4 different events. Shouldn’t it say NA, such as in the positive control?

This was an oversight by the authors.  Thank you for recognizing. Figure 5 has been corrected.

Legend to Fig. 4: The common name of the insect species is written in different styles, with capital letter the first word, the two words, or both in small letters. Please use the correct one and normalize the nomenclature.

The authors completely agree with this sentiment.  However, the importance of genus and species is thought to be most important for this manuscript where global readership may have different common names for Armyworms, in particular.  The Authors previously attempted to use this nomenclature for Figure 4, but the resulting image was too cluttered and unclear with regard to spacing etc.  Therefore, while it is not absolutely ideal, the authors would like to maintain this figure with full descriptions and definitions in the legend.  We have made minor modifications to make capitalization more consistent as suggested.

Fig. 5: It is not clear the meaning of columns Protein ng/cm2 and ng/ul TSP. Also, spell out Na and TSP in the legend.

We modified the legend to now include descriptions of each of these notations.

L 204: Substitute (2018) by [17]

We have corrected this notation.

L 207: The proteins in Banyuls et al. were not chimeric, but Ala substitution mutants. Change chimeric by mutant.

We have corrected this error.

The Discussion section does not mention the interesting, and unexpected, effect of Vip3Ab on the co-feeding experiments. This deserves some comments. The explanation given in Conclusions (lines246-248) is very ambiguous.

The authors agree with this statement and have modified the text as suggested. The co-feeding study is somewhat unexpected and was intriguing to us as well.  However, the singular purpose of this experiment was to investigate the possibility that Vip3Ab1 could “compete” with Vip3Ab1-740.  We obtained a clear negative answer to this question but observed unexpected (and small activity) of Vip3Ab1 in co-dosed insects.  While we would like to understand this better, we are deliberate in avoiding and discussion of synergism because the experiments are not designed appropriately for definitive conclusions regarding this possibility.

L302-303: Why Vip3Ai was expressed if nothing was done with it?

Good question that is now clarified within the text.  The protein was indeed expressed, but not purified from Pseudomonas, because whole cell overlay assays indicated lack of insecticidal activity.

L 307: Define TCEP. Is it a protease inhibitor?

Good question that is now clarified.  TCEP is tris(2-carboxyethyl)phosphine, is a reductant that is a bit more stable and less odorous that DTT.

L 347: Is the title of the heading correct? It has no sense.

This error is now corrected.  Thank you

L 350: Supplemental Table 2 is not a table, but a figure.

This has been corrected.

L 371: The most widely used nomenclature for quantitative PCR is not QPCR, but qPCR. Please change.

These have been changed as requested.

L 389: Capitalize Western.

This is now corrected.

L 397: Dithiothreitol has to go with small letter.

This is now corrected.

L 410: Correct wavelength.

This is now corrected.

Reviewer 2 Report

Also attached.

Re: Review of manuscript entitled: Modification of Vip3Ab1 C-terminus confers broadened plant protection from Lepidopteran pests.

General comments:

The manuscript describes the engineering of a modified synthetic Vip3 protein for the control of Lepidopteran pests.  The manuscript is straightforward and easy to follow.  The methods are well described and appropriate for the study.  The findings are novel and warrant publication in Toxins.  Only minor edits are required before publication, as listed below.

1.      The genus and species names are not consistently italicized. 

2.      On Figure 3, it should be noted that the x-axis for the Vip3Ab1+VipAb1-740 combination refers to the concentration of the modified toxin.  Also, the titles for the toxins on the figure are not consistent.  VipAb1 is on the figure which should read Vip3Ab1, etc.

3.      Include legends for the tables in the methods on pg 10 and 12.

4.      For the insect diet overlay bioassays, I assume that control treatments were used for each treatment.  Report the control mortality or lack there of. 

5.      Have the document edited for minor grammatical errors by a native English speaker. 

Author Response

Thank you for your helpful thoughts and suggestions.  The authors have addressed each point in stepwise fashion below.

General comments:

The manuscript describes the engineering of a modified synthetic Vip3 protein for the control of Lepidopteran pests.  The manuscript is straightforward and easy to follow.  The methods are well described and appropriate for the study.  The findings are novel and warrant publication in Toxins.  Only minor edits are required before publication, as listed below.

1.     The genus and species names are not consistently italicized. 

Thank you.  We have now corrected this issue.

2.      On Figure 3, it should be noted that the x-axis for the Vip3Ab1+VipAb1-740 combination refers to the concentration of the modified toxin.  Also, the titles for the toxins on the figure are not consistent.  VipAb1 is on the figure which should read Vip3Ab1, etc.

Thank you for identifying this issue.  We have corrected this figure accordingly.

3.      Include legends for the tables in the methods on pg 10 and 12.

We have now included title for these details.  Thank you for this point.

4.      For the insect diet overlay bioassays, I assume that control treatments were used for each treatment.  Report the control mortality or lack there of. 

 Yes, each bioassay is run with a positive control and negative control.  Our in-house standard is to only accept bioassay data when control  is less than 15%, which is somewhat arbitrary but is based on an internal standard of control averages over several years of bioassay. This quality control point is now given in the methods.

5.      Have the document edited for minor grammatical errors by a native English speaker. 

Thank you.  We have done this.

Reviewer 3 Report

This manuscript describes a nice piece of work in which the substitution of an alternative C-terminus adds a new specificitity to Vip3Ab. I feel that the work is worthy of publication in Toxins but certain aspects of the manuscript need to be improved.

1) Line 7 - the term "mode of action" is very much an industry term for different specificity but is misleading for academic scientists. If both proteins form pores in the midgut epithelium then many would consider them to have the same MoA.

2) Line 31 - the numbers given for different Vip3 types is misleading. Better to give the number of distinct classes eg 10 different types of Vip3A toxins

3) Line 62 - How can the authors state that Vip3Ai1 lacks insecticidal activity. What they mean is that no activity was observed against the limited range of insects on which it was tested. It should be stated which insects were tested.

4) Line 64 - presumably should say serine protease

5) Line 65 - Why is it worth noting that Vip3Ai has changes in this region? It is only worth noting if the authors have attempted to revert the loss of activity by mutation in this region. I suspect that this has been tried but not reported. Unless such data are presented the authors should not speculate on the impact of these differences.

6) Line 82 and in many places elsewhere within the manuscript and references. Species names are not italicized.

7) Line 82 the manuscript states that the toxins were treated with H. zea gut fluids however this reviewer can find no data showing the results of this.

8) Fig 2 an explanation is required as to why there are two bands at A with the same N-terminus in the undigested samples. Is there C-terminal processing during the expression process?

9) Fig 3 is almost impossible for the reader to figure out without having to carefully scrutinize the text and the manuscript. It should be obvious from the figure itself. The concentrations of both proteins should be shown on the figure not just in the M&M

10) Drawing firm conclusions from Fig 3 is dangerous. I presume that only one biological replicate is presented. There is a clear oddity in that the effect of 111ng/cm2 of Vip1Ab1-740 is less than the effect of 37ng/cm2. Furthermore the same effect is seen with the mixture suggesting that the same (diluted) sample was used for both and that perhaps there was an error in the dilution. At lower Vip1Ab1-740 concentrations it does appear that the addition of Vip3Ab1 increases the activity of the hybrid. However at higher concentrations the effect is not so clear, since there is almost complete mortality which would mask any effect. It would also mask assay differences that could explain the increase in mortality seen at lower concentrations. Basically this reviewer has little confidence in the conclusions drawn from these data and would need to see repeat data,

11) Line 143 spelling mistake

12) Line 247 this conclusion can't be drawn based on Fig 3, especially given the data in Table 1. No evidence that it is partially functional.

13) Line 271 subscript

14) References use italics

Author Response

Thank you for your helpful thoughts and suggestions.  We have addressed each point in stepwise fashion below:

Comments and Suggestions for Authors

This manuscript describes a nice piece of work in which the substitution of an alternative C-terminus adds a new specificitity to Vip3Ab. I feel that the work is worthy of publication in Toxins but certain aspects of the manuscript need to be improved.

1) Line 7 - the term "mode of action" is very much an industry term for different specificity but is misleading for academic scientists. If both proteins form pores in the midgut epithelium then many would consider them to have the same MoA.

YES.  The authors completely agree and have struggled mightily with this both internally and externally. We have changed this statement in the abstract to read – “bind unique receptors to exert their insecticidal effects”.

2) Line 31 - the numbers given for different Vip3 types is misleading. Better to give the number of distinct classes eg 10 different types of Vip3A toxins.

The authors are confused by this statement.  We have no intention to mislead the readers, but do understand your point that many of the Vip3A proteins are quite similar. We are attempted to adjust the statement based on tertiary rank only, which differentiates based on less than or equal to 95% identity.

3) Line 62 - How can the authors state that Vip3Ai1 lacks insecticidal activity. What they mean is that no activity was observed against the limited range of insects on which it was tested. It should be stated which insects were tested.

This is a good point.  We have corrected within the text.

4) Line 64 - presumably should say serine protease

Correct.  Thank you, we have changed this.

5) Line 65 - Why is it worth noting that Vip3Ai has changes in this region? It is only worth noting if the authors have attempted to revert the loss of activity by mutation in this region. I suspect that this has been tried but not reported. Unless such data are presented the authors should not speculate on the impact of these differences.

This is an interesting point.  We have not tried this mutation to this point.  We only highlight this change because the KVKK site is extremely conserved within the Vip3A family and is thought of as a fingerprint.  We have deleted the speculation on the potential impact on toxicity.

6) Line 82 and in many places elsewhere within the manuscript and references. Species names are not italicized.

This is an unfortunate impact of importing to the Toxins format.  We feel we have corrected all of these formatting mistakes.

7) Line 82 the manuscript states that the toxins were treated with H. zea gut fluids however this reviewer can find no data showing the results of this.

This is a good observation.  We did do the exact same experiment with H.zea gut fluids with the identical results.  However, the gel lanes were in slightly different order and it was thought to add little value.  We changed the text of the Figure 2 legend to highlight this point.

8) Fig 2 an explanation is required as to why there are two bands at A with the same N-terminus in the undigested samples. Is there C-terminal processing during the expression process?

This has drawn attention before for us and other groups (Zack et al SciRep, 2017, Kunthic et al BBA 2017).  This an artifact of PAGE analysis of Vip3 proteins in general.  For Edman sequencing and size exclusion, we use a higher concentration of very pure protein (150ug/ml at >90% purity).  Upon exposure to LDS sample buffer (pH ~10.5-11.0), there is some aggregation we assume to be due to alkaline pH as this effect is more pronounced in pH 10 buffer.  We have also performed intact mass analysis of several protein productions and not identified any C-terminal truncations upon expression.

9) Fig 3 is almost impossible for the reader to figure out without having to carefully scrutinize the text and the manuscript. It should be obvious from the figure itself. The concentrations of both proteins should be shown on the figure not just in the M&M.

We agree.  Others reviewers have had similar comments and we have made revisions to this Figure to clarify concentrations and process.

10) Drawing firm conclusions from Fig 3 is dangerous. I presume that only one biological replicate is presented. There is a clear oddity in that the effect of 111ng/cm2 of Vip1Ab1-740 is less than the effect of 37ng/cm2. Furthermore the same effect is seen with the mixture suggesting that the same (diluted) sample was used for both and that perhaps there was an error in the dilution. At lower Vip1Ab1-740 concentrations it does appear that the addition of Vip3Ab1 increases the activity of the hybrid. However at higher concentrations the effect is not so clear, since there is almost complete mortality which would mask any effect. It would also mask assay differences that could explain the increase in mortality seen at lower concentrations. Basically this reviewer has little confidence in the conclusions drawn from these data and would need to see repeat data,

The authors completely understand the reviewer’s view on this – and agree.  We have attempted be brief and concise statement regarding the intention of this experiment as we were not attempting to inspect or draw conclusions beyond competition (Can Vip3Ab1 compete with the chimera in vivo?).  However, as with many experiments, the results were conclusive (no competition, we hope the reviewers agree) but did suggest a possible additive effect. While we would like to understand this better, we are deliberate in avoiding and discussion of synergism because the experiments are not designed appropriately for definitive conclusion regarding this possibility.

At the same time, the authors felt that the results did suggest at least some additive effect that should be mentioned.  We did know this would take on some risk in mentioning in the text for the precise reasons provided.  In fact other reviewers have requested that we provide more discussion on the high mortality observed with co-dosing, for which we have made an argument to decline.   Therefore, we have changed the text to use more direct language highlighting this point.

11) Line 143 spelling mistake

Thank you for identifying.  We have corrected this mistake.

12) Line 247 this conclusion can't be drawn based on Fig 3, especially given the data in Table 1. No evidence that it is partially functional.

We have changed the language to reinforce that conclusion we have made from this experiment, which is that Vip3Ab1 does not compete with Vip3Ab1-740 in the insect midgut to attenuate insecticidal activity.

13) Line 271 subscript

We have corrected this oversight.

14) References use italics

We have corrected this oversight.

Reviewer 4 Report

The work is focused on a chimeric form  of Vip3 protein, named Vip3Ab1-740, and its role against  Spodoptera eridani. Interestingly, transgenic plants expressing Vip3Ab1-740 are protected against several pests. This interesting study could open new perspective on the exploitation of this protein as  broaden insecticidal spectrum.

In general, the manuscript falls within the scope of the journal and the described results add knowledge in the field of plant protection against pests. The experimental work is well performed and the paper is well-written. In particular, the description of results, the discussion and conclusions are convincing and well balanced. 

Please to address the following points:

Line 23: Vegetative insectidical  proteins should be abbreviated as VIPs. Same everywhere of the text.

Line 126: better to explain which regulatory elements have been chosen.

Fig.4: a statistical analysis could be important to assess the efficacy of the recombinant proteins compared to the wild-type in reducing the leaf damage. Please provide the results of that.

Author Response

Thank you for your helpful comments and insight.  We have addressed each point in stepwise fashion below:

Comments and Suggestions for Authors

The work is focused on a chimeric form  of Vip3 protein, named Vip3Ab1-740, and its role against  Spodoptera eridani. Interestingly, transgenic plants expressing Vip3Ab1-740 are protected against several pests. This interesting study could open new perspective on the exploitation of this protein as  broaden insecticidal spectrum.

In general, the manuscript falls within the scope of the journal and the described results add knowledge in the field of plant protection against pests. The experimental work is well performed and the paper is well-written. In particular, the description of results, the discussion and conclusions are convincing and well balanced. 

Please to address the following points:

Line 23: Vegetative insectidical  proteins should be abbreviated as VIPs. Same everywhere of the text.

The authors understand this comment completely.  The evolution to a common nomenclature has been rather slow over the past 20 years.  While there are likely benefits to each method, the authors have observed that the method used “Vip3” is the most common means to describe these proteins. This is likely due to the fact that other vegetative insecticidal proteins (Vip1, Vip2, and Vip4) are completely unrelated in sequence and function (however some are insecticidal and produced in the vegetative growth phase).  We hope the reviewer will understand our aim to be as concise as possible and focus on the Vip3 family.

Line 126: better to explain which regulatory elements have been chosen.

We have added this detail within the text.

Fig.4: a statistical analysis could be important to assess the efficacy of the recombinant proteins compared to the wild-type in reducing the leaf damage. Please provide the results of that.

The authors agree that full statistical breakdown of the experiments similar to this is not uncommon.  In this figure we did not add statistical analysis because of the marked gap between the error bars of the negative controls (WT) and the transgenic plants.  From a readability perspective, this Figure contains a lot of information (scoring, regulatory elements, protein expression levels, etc) and the authors feel that inclusion of comparison bars and associated p values make this figure too busy.  The manuscript does contain a scoring table (which indicates that 0.1 is the lowest score possible) as well as photographs in Figure 5 that show complete protection of transgenic leaf discs.  We hope the reviewers understand our perspective on this point.

Round 2

Reviewer 1 Report

The modification addressed all my queries

Reviewer 3 Report

All previous concerns will be addressed